# Neoadjuvant Treatment for Triple Negative Breast Cancer: Recent Progresses and Challenges

**DOI:** 10.3390/cancers12061404

**Published:** 2020-05-29

**Authors:** Jin Sun Lee, Susan E. Yost, Yuan Yuan

**Affiliations:** Department of Medical Oncology & Molecular Therapeutics, City of Hope Comprehensive Cancer Center and Beckman Research Institute, Duarte, CA 91010, USA; jbitar@coh.org (J.S.L.); suyost@coh.org (S.E.Y.)

**Keywords:** Triple negative breast cancer, neoadjuvant treatment, targeted treatment

## Abstract

Triple negative breast cancer (TNBC) is an aggressive breast cancer with historically poor outcomes, primarily due to the lack of effective targeted therapies. The tumor molecular heterogeneity of TNBC has been well recognized, yet molecular subtype driven therapy remains lacking. While neoadjuvant anthracycline and taxane-based chemotherapy remains the standard of care for early stage TNBC, the optimal chemotherapy regimen is debatable. The addition of carboplatin to anthracycline, cyclophosphamide, and taxane (ACT) regimen is associated with improved complete pathologic response (pCR). Immune checkpoint inhibitor (ICI) combinations significantly increase pCR in TNBC. Increased tumor infiltrating lymphocyte (TILs) or the presence of DNA repair deficiency (DRD) mutation is associated with increased pCR. Other targets, such as poly-ADP-ribosyl polymerase inhibitors (PARPi) and Phosphatidylinositol-3-kinase/Protein Kinase B/mammalian target of rapamycin (PI3K-AKT-mTOR) pathway inhibitors, are being evaluated in the neoadjuvant setting. This review examines recent progress in neoadjuvant therapy of TNBC, including platinum, ICI, PARPi, phosphatidylinositol-4,5-bisphosphate 3-kinase catalytic subunit alpha (PIK3CA) pathway targeted therapies, and novel tumor microenvironment (TME) targeted therapy, in addition to biomarkers for the prediction of pCR.

## 1. Introduction

Triple negative breast cancer (TNBC) accounts for 15% of all breast cancer and it is characterized by the lack of expression of estrogen receptor (ER)/progesterone receptor (PR)/human epidermal growth factor receptor-2 (HER-2), earlier recurrence, tendency of visceral metastasis, and worse overall survival [1,2,3]. The mainstay of treatment for early stage TNBC is neoadjuvant chemotherapy, followed by definitive surgery. Response to initial chemotherapy predicts clinical outcomes in breast cancer [4,5,6]. Neoadjuvant therapy has become increasingly used for the treatment of tumor ≥2 cm in standard-of-care clinical practice, and pathological response is routinely assessed for the evaluation of overall prognosis. Pathological complete response (pCR) was associated with better prognosis in neoadjuvant TNBC trials and has become a surrogate marker of survival [7,8]. The prognosis of TNBC is poor, particularly when pCR was not achieved [9].

Conventional neoadjuvant chemotherapy regimen composed of adriamycin, cyclophosphamide, and paclitaxel (ACT) results in a pCR rate of 35–45% [8,10,11]. Tumor heterogeneity of TNBC is well recognized [12,13,14], yet molecular subtype driven therapy has not become a routine clinical practice, largely due to the lack of effective targeted therapies. Recent clinical trials incorporating immune checkpoint inhibitors (ICI) or targeted therapy, such as poly-ADP-ribosyl polymerase inhibitors (PARPi), may have the potential to personalize neoadjuvant therapy in early stage TNBC [15,16,17]. Other agents, including the phosphatidylinositol-4,5-bisphosphate 3-kinase catalytic subunit alpha (PIK3CA) pathway inhibitors and androgen receptor (AR) targeted therapies, may be applicable for certain molecular subtypes of TNBCs. Novel approaches in conducting neoadjuvant clinical trials, such as I-SPY 2 and ARTEMIS, may accelerate the progress to bring effective targeted therapies to the neoadjuvant setting [18,19,20,21]. In this review, we will summarize recent neoadjuvant trials focusing on the following perspectives: (a) chemotherapy optimization with the addition of carboplatin; (b) the addition of ICI to chemotherapy backbones; (c) clinical trial design by the evaluation of novel targeted agents such as I-SPY 2; (d) biomarker driven identification of clinically relevant patient subgroups to enable a more precise treatment approach (ARTEMIS). Many promising targeted therapies and approaches that are discussed in this review may lead to a paradigm shift of neoadjuvant therapies for early stage TNBC.

## 2. Molecular Heterogeneity of Triple Negative Breast Cancer

A subset of TNBC is chemosensitive and 35–45% of patients achieve pCR despite the poor prognosis and aggressive nature. This might be explained by the tumor molecular heterogeneity of TNBC. Several molecular classifiers that are based on mRNA profiling of TNBC have been identified. Lehmann et al. reported TNBC-7 subtypes [12]: basal-like 1 (BL1), basal-like 2 (BL2), immunomodulatory (IM), mesenchymal (M), mesenchymal stem-like (MSL), luminal androgen receptor (LAR), and unstable (UNS). The BL1 subtype expresses genes that are related to cell cycle and proliferation, which is consistent with the elevated DNA damage response pathway observed in this subtype. This molecular property explains the robust response to neoadjuvant chemotherapy of the BL1 subtype [14], specifically sensitivity to platinum agents [22]. The BL2 subtype expresses genes that are associated with growth factor signaling. The IM subtype highly expresses immune cell signaling genes. The M and MSL subtypes express genes that are involved in cell motility, and the MSL subtype is also associated with cell differentiation. The LAR subtype demonstrates distinctive gene expression that is enriched in hormone-regulated pathways, such as AR signaling, and it is the least proliferative subtype resulting in enhanced chemotherapy resistance. The clinical relevancy of the TNBC 7-subtype was further investigated by determining pCR rates after neoadjuvant chemotherapy. Of 146 patients with TNBC, molecular subtype and pCR status were significantly associated (*p* = 0.04379) and TNBC subtype was an independent predictor of pCR status (*p* = 0.022) by a likelihood ratio test [23]. The BL1 subtype had the highest pCR rate (52%); BL2 and LAR had the lowest (0% and 10%, respectively). Similarly, in a study conducted by Santonja et al., 125 TNBC patients treated with neoadjuvant anthracyclines and/or taxanes +/− carboplatin showed BL1 tumors had the highest pCR to carboplatin containing regimens (80% vs. 23%, *p* = 0.027) and LAR tumors had the lowest pCR to all treatments (14.3% vs. 42.7%, *p* = 0.045 when excluding MSL samples) [22]. Later, these seven subtypes were refined into four types (TNBC type-4): BL1, BL2, M, and LAR with evidence of IM and MSL subtypes representing tumors with substantial infiltrating lymphocytes and mesenchymal cells, respectively. The BL1 subtype demonstrated the highest pCR rate of 40–50% [14]. 

Burstein et al. subdivided TNBCs into TNBC-4 subtypes: LAR, mesenchymal (MES), basal-like immunosuppressed (BLIS), and basal-like immune-activated (BLIA) [13]. The LAR subtype demonstrated molecular evidence of ER activation suggesting response to anti-estrogen or anti-androgen therapies, as described in Lehmann’s subtypes [12]. MES subtype was characterized by pathways of cell cycle, mismatch repair, and DNA damage repair. The BLIS subtype exhibited a downregulation of immune and cytokine pathways that are associated with the worst clinical outcomes. Contrary to BLIS, the BLIA subtype showed the best clinical outcomes with upregulated immunoregulation pathways. 

Despite the fact that TNBC subtyping provides an in-depth understanding of the tumor heterogeneity of TNBC [24,25,26], its clinical application has been limited due to the complexity of gene signatures. Table 1 summarizes molecular subtypes of TNBC and potential targets for therapies.

## 3. Platinum in TNBC

Platinum salts are DNA damaging agents that show an increased efficacy in the tumors with a defected DNA repair system. The platinum salts react with DNA inside cells and distort the double helix of DNA inducing single-strand breaks (SSB) and double-strand breaks (DSB). When these damages cannot be efficiently repaired, it results in cell death [28]. These agents have shown activity in cancers with germline BRCA mutation, as BRCA 1/2 proteins have an essential role in repairing DNA damage [29,30,31].

A high proportion of TNBC exhibits BRCA-like status (BRCAness), which indicates that these tumors are highly sensitive to platinum salts [32,33]. Platinum-based chemotherapy has been investigated in the neoadjuvant setting, with the goal to increase pCR and improve clinical outcome. Carboplatin-containing regimens demonstrated superior pCR rates when compared with standard regimen in two large randomized trials. The CALGB 40603/Alliance trial studied the clinical benefit of adding carboplatin +/− bevacizumab to neoadjuvant chemotherapy in stage II/III TNBC [34]. A total of 443 patients with stage II/ III TNBC received paclitaxel 80 mg/m^2^ weekly × 12, followed by dose dense doxorubicin and cyclophosphamide (ddAC) × 4, and were randomly assigned to concurrent carboplatin (AUC 6) once every three weeks × 4 ± bevacizumab 10 mg/kg every two weeks × 9. With carboplatin, the percentage of patients who achieved pCR increased significantly from 41% to 54% (odds ratio (OR), 1.71; *p* = 0.0029). The trial was not powered to detect long term overall survival (OS) and the addition of carboplatin to standard chemotherapy did not improve long term OS [35]. In the GeparSixto trial, 595 patients with stage II and III TNBC were randomized to receive either carboplatin or no carboplatin on a backbone regimen with paclitaxel, liposomal doxorubicin, and bevacizumab [36]. The pCR rates were significantly improved in the carboplatin group: 53.2% vs. 36.9 (*p* = 0.005). In both the CALGB 40603 and GeparSixto trials, hematological toxicities, including neutropenia and thrombocytopenia, were increased in the carboplatin group. The result from a meta-analysis of nine randomized controlled trials (RCTs) (N =  2109) showed that platinum-based neoadjuvant chemotherapy significantly increased pCR rate from 37.0% to 52.1% (OR 1.96, 95% confidential interval (CI) 1.46–2.62, *p* < 0.001) [37]. In addition, an increased pCR rate persisted after restricting the analysis to the three RCTs (N = 611) that used the same standard regimen in both groups of weekly paclitaxel (with or without carboplatin), followed by doxorubicin and cyclophosphamide (AC) (OR 2.53, 95% CI 1.37–4.66, *p* = 0.003). In two of the RCTs (N = 748) with survival data reported, no significant difference in event free survival (EFS) (hazard ratio (HR) 0.72, 95% CI 0.49–1.06, *p* = 0.094) and OS (HR 0.86, 95% CI 0.46–1.63, *p* = 0.651) were observed. Significantly increased grade 3/4 hematological adverse events (AEs) were observed with platinum-based neoadjuvant chemotherapy. 

Our single center phase II trial of carboplatin plus nab-paclitaxel (carboplatin AUC6 every four weeks × 4 and weekly nab-paclitaxel at 100 mg/m^2^ × 16 week) in stage II-III TNBC (N = 67) demonstrated a pCR rate of 48% with reasonable tolerability [38]. Sharma et al. reported a pCR rate of 55% with the combination of carboplatin and docetaxel (carboplatin AUC6, docetaxel 75 mg/m^2^ every three weeks × 6, N = 190) [39]. Table 2 shows the pCR rates from clinical trials that explored the efficacy of carboplatin.

In addition to studies with carboplatin, cisplatin 75 mg/m^2^, every three weeks × 4 was evaluated in a randomized phase II study of neoadjuvant cisplatin vs. AC in germline BRCA carriers with HER2-negative (TBCRC 031) [46]. The pCR rate was 18% with cisplatin and 26% with AC, yielding a risk ratio (RR) of 0.70 (90% CI, 0.39 to 1.20).

Other chemotherapy agents were evaluated to improve clinical outcomes in TNBC without compelling results. Nab-paclitaxel, an albumin-bound particle form of paclitaxel, has shown preferential tumor uptake and favorable safety profiles when compared to paclitaxel [47,48,49]. The clinical benefit from nab-paclitaxel in TNBC is still controversial [50,51]. When nab-paclitaxel was used instead of paclitaxel in phase III trial, the improvement of pCR was not statistically significant, showing pCR 41.3% with nab-paclitaxel vs. 37.7% with paclitaxel in the TNBC group (OR 0.85; 90% CI, 0.49–1.45) [52]. The GeparSepto trial that included about 20% of TNBC demonstrated significantly higher pCR in nab-paclitaxel subgroup than in the paclitaxel subgroup (48% vs. 26%, *p* = 0.00027) [53]. Additionally, nab-paclitaxel showed superior pCR when given with carboplatin as compared with gemcitabine [44].

## 4. Immune Check Point Inhibitors 

Introducing immunotherapy in the oncology field has changed the landscape of cancer treatment. Programmed death-1 (PD-1) is a T-cell inhibitory receptor that regulates the immune system by downregulating T-cell response upon binding with its ligand, programmed death ligand-1 (PD-L1) expressed on cancer cells. While the activation of this pathway prevents cancer cells from immune mediated cell death, the inhibition of PD-1 or PD-L1 can restore anti-tumor effects of T-cells. Among all of the breast cancer subtypes, TNBC is especially immunogenic, exhibiting increased expression of PD-L1 [54,55]. This immunogenicity observed in TNBC attracted ICI as a treatment option. 

PD-L1 inhibitor atezolizumab showed progression free survival (PFS) benefit in metastatic TNBC (mTNBC) and a complete response (CR) of 10%, which lead to the first ICI approval in TNBC [56]. ICIs have been investigated in several neoadjuvant trials in TNBC. The primary objective of these trials is to test pCR from adding ICI to chemotherapy, mainly taxane and anthracycline. Notably, carboplatin has been utilized when considering its efficacy in TNBC based on previous trials. KEYNOTE-173 is a phase Ib study that showed improved pCR rate from programmed cell-death 1 (PD-1) inhibitor pembrolizumab combined with neoadjuvant chemotherapy [16]. KEYNOTE-522 is a phase III study of neoadjuvant chemotherapy (paclitaxel and carboplatin, then doxorubicin or epirubicin and cyclophosphamide) combined with pembrolizumab or placebo, followed by adjuvant pembrolizumab or placebo in patients with TNBC [15]. An interim analysis reported significantly higher pCR rate in the pembrolizumab combined group (64.8% vs. 51.2%) than in the chemotherapy alone group, regardless of PD-L1 status. Event-free survival (EFS) was significantly higher in the pembrolizumab group during median follow up of 15.5 months. Grade 3 or higher AEs were 76.8% and 72.2%, respectively, with neutropenia as the most common serious AEs in both groups. This is the first phase III trial supporting the role of ICI in neoadjuvant and adjuvant treatment, and a long-term survival result is expected. In I-SPY 2, the overall pCR rate reached 60% when pembrolizumab was given with paclitaxel followed by AC [57].

Durvalumab has also been evaluated for neoadjuvant treatment. In a phase I/II trial, durvalumab with nab-paclitaxel and sequential ddAC carried 55% of pCR in the PD-L1 positive group [58]. GeparNuevo trial randomized patients with TNBC to nab-paclitaxel with or without durvalumab [59]. All of the patients then received epirubicin and cyclophosphamide (EC) as neoadjuvant chemotherapy. Among 174 patients, the pCR rate with durvalumab was 53.4% vs. placebo 44.2%. Interestingly, this increased pCR rate was seen exclusively in patients that were treated with durvalumab alone before the initiation of chemotherapy (pCR 61%). In the NeoTRIP study (NCT002620280), 280 patients with TNBC were randomized to receive neoadjuvant carboplatin AUC 2 and nab-paclitaxel at 125 mg/m^2^ intravenously (IV) on days 1 and 8 with or without atezolizumab at 1200 mg intravenously on day 1 [60]. The pCR rate was 43.5% (95% CI, 35.1%–52.2%) with atezolizumab and 40.8% (95% CI, 32.7%–49.4%) without atezolizumab in the intent-to-treat population, which led to an odds ratio of 1.11 (95% CI, 0.69–1.79; *p* = 0.066). In this study, 49% of patients had cT2 disease, 59% had cN1 nodal status, and 56% were PD-L1 positive. The role of atezolizumab in the neoadjuvant setting is currently being investigated in the GeparDouze/NSABP B-59 (NCT03281954) trial. TNBC patients will receive neoadjuvant atezolizumab combined with chemotherapy (carboplatin plus paclitaxel and AC or EC), followed by adjuvant atezolizumab [61]. Clinical trials evaluating ICIs in early stage TNBC are described in Table 3.

Based on these encouraging pCR rates, ICIs may play an important role in neoadjuvant therapy for TNBC and eventually become standard-of-care for a subset of TNBCs. However, well defined biomarkers for the better identification of appropriate patients remain lacking. Long-term survival benefits from adding ICIs need to be evaluated in order to adopt ICIs as neoadjuvant treatment in clinical practice. 

## 5. TME Targeting for Neoadjuvant Treatment

The tumor microenvironment (TME) is associated with immune suppression, escape from immune detection and development of drug resistance, and is being increasingly recognized as a potential target for treatment of TNBC [63]. Tumor associated macrophages (TAMs) promote the progression and metastasis of TNBC by releasing inhibitory cytokines, reducing functions of tumor infiltrating lymphocytes (TILs), promoting T_REG_ (regulatory T-cells), and modulating PD-1/PD-L1 expression in TME [64]. TME targeted therapies are undergoing active clinical trial investigation. The combination of cabiralizumab, an antibody that inhibits the colony stimulating factor-1 receptor (CSF1R) and it blocks the activation and survival of macrophages, and ICI with neoadjuvant chemotherapy might improve efficacy by decreasing TAMs and increasing TILs in early stage TNBC [65,66]. Currently, cabiralizumab is being used in combination with nivolumab and neoadjuvant chemotherapy in patients with localized TNBC (NCT04331067).

## 6. PARP Inhibitors for Neoadjuvant Treatment

DNA repair deficiency and PARP inhibitors: BRCA 1/2 mutation is one of the greatest genetic risk factors of developing breast cancer. BRCA 1 and BRCA2 are tumor suppressor genes that play a major role in the DNA repair system, specifically in homologous recombination, which repairs double-stranded breaks (DSBs). When homologous recombination does not function (homologous recombination deficiency, HRD), commonly seen in cases of BRCA 1/2 mutations, DSBs result in genomic instability. Poly ADP-ribose polymerase (PARP) 1 is a protein that binds to single stranded breaks (SSBs) during the DNA repair process. PARPi traps PARP1 and induces cell death by preventing SSB repair, followed by DSBs without functional homologous recombination in patients with BRCA mutations. PARPi showed efficacy in patients with BRCA mutations. The OlympiAD trial is a randomized phase III trial that compared olaparib with standard chemotherapy in patients with metastatic breast cancer and germline BRCA mutation [67]. The significantly longer PFS shown in the olaparib group was more prominent in the TNBC (HR 0.43 in TNBC vs. 0.82 in non-TNBC) and made up of 50% of this study. Talazoparib is aPARPi that is approved for advanced breast cancer with BRCA mutation through the EMBRCA trial [68]. In this trial, patients with germline BRCA mutation were randomized in a 2:1 ratio to receive talazoparib or single-agent therapy of the physician’s choice. Among a total of 287 patients who received talazoparib, 45% patients were TNBC. The median PFS was significantly longer in the talazoparib group. 

The efficacy of PARPi was further investigated in the neoadjuvant setting. I-SPY 2 trial studied the neoadjuvant PARPi veliparib and carboplatin followed by AC as compared with standard neoadjuvant chemotherapy (paclitaxel followed by AC) [69]. The estimated pCR rate in TNBC was 51% in the veliparib-carboplatin group vs. 26% in the control group. This study demonstrates that patients with TNBC can benefit from PARPi and carboplatin as neoadjuvant treatment. However, the grade 3 or 4 hematologic toxicity was much higher in the veliparib-carboplatin group than in control group. BrighTNess trial is a phase III randomized trial to confirm clinical benefit of adding veliparib and carboplatin in TNBC [45]. A total of 634 patients with stage II-III TNBC were randomized (2:1:1) to receive veliparib/carboplatin/paclitaxel, carboplatin/paclitaxel, or paclitaxel, followed by AC after the randomized portion. The pCR rate in veliparib/carboplatin/paclitaxel group (53%) was significantly higher than the paclitaxel group (31%), but adding veliparib to carboplatin/paclitaxel did not improve pCR (58%). Grade 3–4 hematology toxicity was significantly increased from adding carboplatin, regardless of using veliparib. 

Several studies have been conducted to evaluate the role of PARPi as neoadjuvant treatment in early stage BRCA mutated or HRD breast cancer. Most patients enrolled in these studies are TNBC. MD Anderson reported a pilot study of neoadjuvant talazoparib in patients with germline BRCA mutations [70]. Fifteen of 20 enrolled patients were TNBC, and 50% achieved pCR after six months of single agent talazoparib. Hematologic toxicity was the most common AE with 40% Grade 3 anemia and 15% Grade 3 neutropenia. This enhanced pCR from single agent talazoparib offers a different approach for patients with early stage BRCA mutated TNBC. A confirmatory trial is currently ongoing in order to verify the benefit of single agent talazoparib (NCT02401347). GeparOLA trial (NCT02789332) is a phase II randomized trial to evaluate neoadjuvant paclitaxel and olaparib in patients with HRD [71]. The study randomized 107 patients, including 77 TNBC, to either weekly paclitaxel and daily olaparib or weekly paclitaxel and carboplatin for 12 weeks, and then followed with EC. Interestingly, an improved pCR rate in the olaparib group was achieved in the hormone receptor-positive group (29 patients). In TNBC, the olaparib group showed a pCR rate of 56% and the carboplatin group showed 59.3%. Carboplatin is a DNA damaging agent, and works in a similar way to PARPi by inducing DNA damage through DSBs in HDR resulting in a comparable pCR. Table 4 summarizes clinical trials utilizing PAPRi. 

The benefit of adding PARPi to neoadjuvant chemotherapy is still controversial for all TNBCs, despite its correlation with genomic instability. The improved pCR from adding carboplatin (also a DNA breaking agent) makes the use of PARPi more controversial. Large randomized trials are needed to determine whether adding carboplatin, PARPi, or both can achieve better pCR. Importantly, the toxicities from adding carboplatin or PARPi should be considered as increased hematologic AEs were observed in previous trials.

## 7. PI3K/AKT/mTOR Targeted Therapies 

Phosphatidylinositol-3-kinase (PI3K)/AKT/ mammalian target of rapamycin (mTOR) signaling is the most commonly activated cancer driver pathway, leading to cell proliferation and survival (Figure 1). The mutation of *PIK3CA*, the gene encoding the subunit p110α of PI3K, or deactivation of phosphatase and tensin homolog (PTEN), negative regulator of PI3K, can contribute to the progression of cancer [72,73,74]. *PIK3CA* mutation is found in 20–40% of breast cancer and it is associated with increased resistance to chemotherapy [75,76,77]. The inhibition of this pathway has been actively investigated [78], and the mTOR inhibitor everolimus and the PI3K-α inhibitor alpelisib were FDA approved for hormone receptor-positive metastatic breast cancer [79,80]. The alterations of the PI3K/PTEN/AKT pathway (including PIK3CA mutations, PTEN inactivating mutations, and AKT1 activating mutations) occur in 25% of primary TNBC and possibly at a modestly higher frequency in mTNBC [81,82,83]. In the phase II LOTUS trial, the patients were randomly assigned (1:1) to receive intravenous paclitaxel 80 mg/m^2^ (days 1, 8, 15) with either ipatasertib, a pan-AKT inhibitor at 400 mg or placebo once per day (days 1–21) every 28 days. Median PFS in the intention-to-treat (ITT) population was 6.2 months (95% CI 3.8–9.0) with ipatasertib versus 4.9 months (3.6–5.4) with placebo (HR 0.60, 95% CI 0.37–0.98; *p* = 0.037). In the 48 patients with PTEN-low tumors, the median PFS was 6.2 months (95% CI 3.6–9.1) with ipatasertib vs. 3.7 months (range 1.9–7.3 months) with placebo (HR 0.59, 95% CI 0.26–1.32, *p* = 0.18) [84]. The Phase III IPATUNITY130 trial (NCT03337724), where patient receive either paclitaxel and ipatasertib or paclitaxel and placebo, will confirm survival benefit [85]. The AKT inhibitor Capivasertib showed significantly longer PFS and OS in mTNBC when added to paclitaxel as first line treatment of mTNBC [86]. 

Ipatasertib has been studied in neoadjuvant TNBC in a phase II neoadjuvant FAIRLANE study of weekly paclitaxel plus ipatasertib or placebo with the following endpoints: pCR rate, PTEN-low population assessed via IHC and PIK3CA/AKT1/PTEN-altered tumors using next generation sequencing (NGS) [87]. The addition of ipatasertib showed a small increase in pCR rate of 17% vs. 13% in ITT. The clinical response rate by breast MRI of ipatasertib was numerically improved, but not statistically significant in the biomarker-selected patients: PTEN-low tumors (32% vs. 6%) and PIK3CA/AKT1/PTEN-altered tumors (39% vs. 9%). In the adaptive neoadjuvant phase II I-SPY 2 trial, the AKT inhibitor MK-2206 plus standard neoadjuvant chemotherapy of weekly paclitaxel followed by AC achieved an estimated pCR rate of 40% when compared with 22% from chemotherapy alone in the TNBC subgroup [88]. These results support further evaluation of AKT inhibition + paclitaxel and AC neoadjuvant chemotherapy in patients with PIK3CA/AKT1/PTEN-altered tumors.

## 8. Androgen Receptor Targeting in TNBC

AR is a potential therapeutic target considering 10–40% of TNBC express AR of 1 to 10% of stained tumor cells [89,90,91]. The efficacy of AR inhibitors has been studied in AR-positive mTNBC. The AR inhibitor enzalutamide has demonstrated clinical benefit rate at 16 weeks of 33% in mTNBC with AR ≥ 10% [92]. Abiraterone, an inhibitor of 17α-Hydroxylase/C17,20-lyase (CYP17) required enzyme for androgen biosynthesis, had modest objective responsive rate (ORR) of 6.7%, PFS of 2.8 months and six month clinical benefit rate (CBR) of 20% in AR-positive (≥10% IHC) mTNBC [93]. Single agent AR targeted therapy appears to be modest, and combination therapy with other targeted agents are currently under investigation. Enzalutamide plus paclitaxel neoadjuvant therapy is currently ongoing (NCT02689427). Enzalutamide in combination with taselisib (NCT02457910) or alpelisib (03207529) trials in mTNBC are actively accruing patients. 

## 9. Biomarkers Predicting pCR in TNBC 

### 9.1. Tumor Infiltrating Lymphocytes

While immunotherapy is successful across a variety of tumor types, biomarkers precisely predicting response to therapy remain to be identified. Understanding the tumor immune microenvironment holds promise for optimal cancer therapy. TILs and PD-L1 and are the most commonly used biomarkers to evaluate the response to ICI. The presence of stromal tumor infiltrating lymphocytes (sTILs) is widely recognized as a good prognostic factor in both adjuvant and neoadjuvant chemotherapy [94,95,96]. Loi et al. reported that higher levels of TILs were associated with decreased distant recurrent in TNBC, and improved disease free survival (DFS) and OS [97]. Two pooled analyses with a large number of patients demonstrated that increased TILs predict pCR and improved survival in TNBC. The German Breast Cancer Group analyzed pretreatment core biopsies from 3771 patients for sTILs following the guidelines of the International TIL working group [98]. TILs were predefined in three groups: low (0–10%), intermediate (11–59%), and high TILs (≥60%). Increased TIL percentile predicted response to neoadjuvant chemotherapy in TNBC: pCR was achieved in 80/260 (31%) of patients with low TILs, 117/373 (31%) of patients with intermediate TILs, and 136/273 (50%) of patients with high TILs (*p* < 0.0001). A 10% increase in TILs was associated with longer DFS in TNBC (HR 0.93 (95% CI 0·87–0·98), *p* = 0.011) and longer overall survival in TNBC (HR 0·92 (95% CI 0·86–0·99), *p* = 0.032). These findings were reproduced in a different pooled analysis with 2148 patients from nine studies for adjuvant chemotherapy [99]. Mean sTILs was 23% and increased sTILs were significantly associated with improved survival: HR for a 10% increase in sTILs was 0.83 (95% CI, 0.78–0.87) for distant DFS and 0.83 (95% CI, 0.79–0.88) for OS. sTILs significantly decreased in metastatic TNBCs as compared with matched primary [100,101]. Higher TIL PD1 expression was associated with better prognosis in early stage TNBCs [102]. These results further support the approach of introducing ICIs early in the neoadjuvant or adjuvant setting, since primary tumors are more immunogenic. 

### 9.2. PD-L1

PD-L1 expression on tumor cells or immune cells has been evaluated as a biomarker of treatment response to anti-PD-1 or anti-PD-L1 therapies [103,104]. Measuring PD-L1 expression remains controversial due to different methods and antibodies. The expression of PL-L1 in TNBC was estimated to be 40–65% on immune cells [105,106]. Mittendorf et al. reported 19% (20 /105 TNBC) of tumor cells were PD-L1 positive, defined by >5% of membranous staining by IHC [54]. In IMpassion 130 trial, intratumoral CD8 correlated with PD-L1 immune cell expression, and was therefore predictive of prolonged PFS (HR, 0.74; 95% CI, 0.61–0.91) and OS (HR, 0.66; 95% CI, 0.50–0.88) with atezolizumab and nab-paclitaxel vs. placebo and nab-paclitaxel [107]. sTILs were not well correlated with PD-L1 immune cell expression, and only predicted prolonged PFS with atezolizumab when compared with placebo (HR, 0.66; 95% CI, 0.50–0.86). There is a lack of quantitative association between PD-L1 expression and response. Indeed, the response to ICI is not linearly associated with increasing levels of expression, and the methods and antibodies used for PD-L1 assessment remain controversial [108]. It has been observed that PD-L1 negative patients may still derive benefit from ICIs. The knowledge gap in PD-L1 testing across different trials needs to be mitigated in order to best characterize patients who might benefit from ICIs.

### 9.3. Immune Gene Signatures

In addition to TILs and PD-L1, multi-gene signatures have been studied as a more comprehensive tool capturing the immunogenicity of TNBC. The GeparSixto trial was analyzed for mRNA markers from pretherapeutic formalin-fixed paraffine embedded core biopsied samples [109]. A GeparSixto immune signature (GSIS) composed of seven immune-activating genes (*CXCL9*, *CCL5*, *CD8A*, *CD80*, *CXCL13*, *IGKC*, *CD21*) and five immunosuppressive (*IDO1*, *PD-1*, *PD-L1*, *CTLA4*, *FOXP3*) genes was validated as a marker for immune reaction. GSIS revealed that the increased mRNA expression level of these genes, including immunosuppressive genes, was associated with pCR. In our neoadjuvant carboplatin and nab-paclitaxel trial, GSIS was significantly associated with pCR and residual cancer burden (RCB) in a multivariate model (submitted) [38].

### 9.4. Combined Modality of Gene Signatures and IHCs

Using laser capture microdissection gene expression profiles, the tumor immune microenvironment (TIME) was captured and subclassified from therapy-naïve TNBC tumors. An “immune hot” TIME exhibited tumor infiltration of granzyme B^+^CD8^+^ T cells (GzmB^+^CD8^+^ T cells), a type 1 IFN signature, and elevated expression of multiple immune inhibitory molecules, including indoleamine 2,3-dioxygenase (IDO) and PD-L1, was associated with good outcomes. An “immune-cold” TIME with an absence of tumoral CD8^+^ T cells was defined by elevated expression of the immunosuppressive marker B7-H4, signatures of fibrotic stroma, and was associated with poor outcomes [110]. This laboratory approach appears to be labor-intensive and might not be easily adapted in the clinic. 

### 9.5. BRCAness or DNA Repair Defect

Recent advanced technology can capture functional HRD beyond BRCA 1/2 mutations (BRCAness) that shares molecular features of BRCA alteration with a scoring system. myChoice HRD^®^ by Myriad Genetics (Salt Lake City, UT, USA) is a commercially available test for assessing HRD. This is a NGS-based in vitro test that determines genomic instability that is based on an algorithmic scoring system of loss of heterozygosity (LOH), telomeric allelic imbalance (TAI), and large-scale state transitions (LST) [111]. TNBC is strongly related with BRCA mutation and HRD. Among all of the patients with TNBC, 10–15% of patients have germline BRCA 1/2 mutations [112,113] and 40–60% of patients are positive for HRD [114,115,116]. Other genes that are involved in the HR process, such as *PALB2* and *RAD51*, have been discovered to play an important role in TNBC [117]. TNBC with BRCA mutation or HRD is more sensitive to chemotherapy or PARPi [111,118,119,120]. However, evaluating HRD has not been standardized in clinical practice. Currently there are few commercially available methods to evaluate HRD, and this needs to be further studied before being used in clinical practice. 

## 10. Novel Neoadjuvant Clinical Trial Approach

Several ongoing trials have been evaluating the addition of novel targeted therapy agents to standard chemotherapy in the neoadjuvant setting, including I-SPY 2 and ARTEMIS trial. The I-SPY 2 trial utilizes an adaptive design for evaluating the addition of novel agents to paclitaxel, followed by AC (P-AC) in high-risk early stage breast cancer [121]. The addition of veliparib to P-AC had an estimated pCR of 51% [69] and adding pembrolizumab to P-AC had an estimated pCR rate of 60% [57]. Although improved pCR is encouraging, the addition of veliparib or pembrolizumab has not been conclusively shown to improve long-term outcome. This might be attributed to the small sample size.

Precision medicine based on the genomic tests has been adopted in clinical trials. In the metastatic setting, the utility of genomic mutation driven therapies has been tested in basket trials, such as NCI-MATCH (Molecular Analysis for Therapy Choice), which contains a multi-arm design with each arm testing a single drug on a histology-agnostic fashion [122,123]. Despite the appealing concept of precision medicine for management of metastatic breast cancer, the implementation of such approaches in the neoadjuvant setting remains challenging. While TNBC patients with pCR/RCB-0 or RCB-1 have better survival, those with extensive residual disease (RCB-II or RCB-III) after neoadjuvant chemotherapy (NACT) have poor prognoses [124,125,126]. The ARTEMIS (NCT 02276443) is a randomized phase II trial to determine whether precision neoadjuvant therapy (P-NAT) impacts the rates of pathologic response (RCB 0–I) while using a CLIA-certified chemosensitivity mRNA gene signature (GES) and subtyping of TNBC by IHC to select targeted therapy trials for chemotherapy-insensitive tumors [19,20,21]. The initial study plan was to randomize 360 patients with TNBC as 2:1 ratio to “know” vs. “not know” P-NAT. Chemotherapy-sensitive tumors received chemotherapy, and chemotherapy-insensitive tumors were enrolled in a clinical trial. The first interim analysis (N = 133 patients with RCB status) revealed a RCB 0–1 rate of 56% (“know” P-NAT) vs. 62% (“not know” P-NAT); *p* = 1.0; thus, randomization was discontinued for futility [19]. A total of 232 patients were enrolled, including 168 evaluable for RCB. In the ultrasound-resistant cohort (N = 43), RCB 0–I rates were higher in patients treated with targeted therapy (N = 30) vs. AC-T (N = 13); (30% vs. 8%; odds ratio = 5.1 with 95% CI, 0.6–45.7; *p* = 0.11). GES failed to improve the rates of RCB 0–I in TNBC; however, in patients with resistant disease identified by ultrasound after AC, RCB 0–I rates were higher in patients that were treated with targeted therapy as compared to chemotherapy alone. This trial again demonstrated a persistent gap between tumor biology and the clinical application of precision medicine in the neoadjuvant setting.

## 11. De-Escalation vs. Escalation of NAC Regimen in TNBC

The optimization of a neoadjuvant chemotherapy regimen in early stage TNBC continues to evolve. The key question remains to be the appropriate selection of a neoadjuvant regimen based on patient and disease characteristics. Recently, the promising pCR rate that was reported from KEYNOTE 522 using pembrolizumab plus carboplatin/paclitaxel followed by AC potentially shifts the standard-of-care regimen for early stage TNBC neoadjuvant therapy toward more intensive chemotherapy backbone, such as weekly carbo/taxol followed by AC, although significant Grade 3–4 AEs raised the question of whether every patient requires such an intensive regimen. 

There is existing evidence showing the carboplatin/taxane regimen remains highly active and it could serve as a chemotherapy backbone for immunotherapy or targeted therapy combinations (de-escalated chemotherapy). Recent trials demonstrated a promising pCR rate of “anthracycline-free regimen”. Our single center phase II trial of carboplatin plus nab-paclitaxel (carboplatin AUC6 every four weeks × 4 and weekly nab-paclitaxel at 100 mg/m^2^ × 16 week) in stage II-III TNBC (N = 67) demonstrated a pCR rate of 48% with reasonable tolerability [38]. Sharma et al. reported a pCR rate of 55% with the combination of carboplatin and docetaxel (carboplatin AUC6, docetaxel 75 mg/m^2^ every three weeks × 6, N = 190) [39]. WSG-ADAPT-TN trial reported by Gluz et al. also reflected a de-escalation concept with a 12 week neoadjuvant regimen [44]. When patients were treated with carboplatin AUC2 with nab-paclitaxel 125 mg/m^2^ on day 1 and 8 for four three-week cycles, the pCR rate was 44.9 (N = 154). The expression of immunological genes (CD8, PD-L1), basal-like mRNA expression profile, and high Ki-67 were associated with pCR in a multi-variate model (*p* < 0.05) [127]. All three trials are consistent with a favorable toxicity profile and high efficacy using carboplatin and taxane based anthracyclin-free regimen. These data support further research while using de-escalated chemotherapy backbone for combination therapy with ICI or targeted therapy. Currently, the ongoing NeoPACT trial combining carboplatin AUC6, docetaxel 75 mg, and pembrolizumab 200 mg every three weeks is actively enrolling patients, and the results of the trial result are eagerly awaited (NCT03639948). Confirmatory analysis of biomarkers predicting patients who can achieve pCR without the use of anthracycline and/or ICIs is critical for patient selection.

## 12. Conclusions and Future Direction

Recent progress has been made in neoadjuvant therapy for early stage TNBC. ICI and PARPi may become standard-of-care for appropriate subtypes of TNBC. Carboplatin remains an important treatment in BRCA-associated tumors or HRD tumors. Novel clinical trial design, such as I-SPY 2 or ARTEMIS, might vastly facilitate testing novel targeted therapy in the neoadjuvant setting. The many promising targeted therapies and approaches that are discussed in this review may lead to a paradigm shift of neoadjuvant therapies for early stage TNBC. 

## Figures and Tables

**Figure 1 cancers-12-01404-f001:**
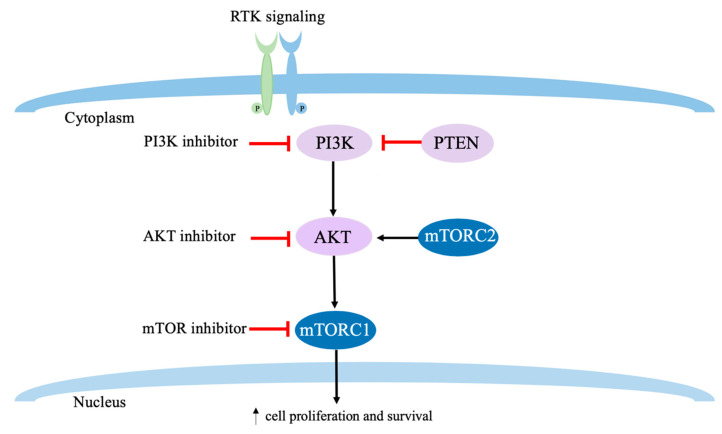
Mechanisms of PI3K/AKT/mTOR pathway activation and targeted therapies. Activating mutations in the α catalytic domain of PI3K and/or *PTEN* mutation lead to pathway activation. PI3K signaling pathway linking RTK signaling leads to downstream activation of PI3K/AKT/mTOR, promoting cell proliferation and survival. *RTK* receptor tyrosine kinase, PI3K phosphatidylinositol-3-kinase, *PTEN* phosphatase and tensin homolog, AKT Protein Kinase B, *mTORC* mechanistic target of rapamycin complex.

**Table 1 cancers-12-01404-t001:** Triple negative breast cancer (TNBC) molecular subtype and potential targets for therapy.

Molecular Subtypes	Genomic Alterations	Potential Therapeutic Targets
Basal-like 1 (BL1)	Cell cycle DNA repair (ATR–BRCA pathway) Proliferation	PARP inhibitors Carboplatin, Cisplatin Other chemotherapy
Basal-like 2 (BL2)	Growth factor signaling pathways (EGFR, MET, NGF, Wnt/β-catenin, IGF-1R) Glycolysis, gluconeogenesis Expression of myoepithelial markers	mTOR inhibitors Growth-factor inhibitors
Immunomodulatory (IM)	Immune cell processes (CTLA4, IL12, IL7 pathways, antigen processing/presentation) Gene signature for medullary BC (rare TNBC with a favorable prognosis)	PD1/PD-L1 inhibitors Other immune checkpoint inhibitors
Mesenchymal-like (M)	Cell motility Cell differentiation Growth factor signaling (*Notch*, PDGFR, FGFR, TGFβ) EMT	mTOR inhibitors EMT-targeted therapy CSC-targeted therapy AXL inhibitor
Mesenchymal stem-like (MSL)	Low proliferation Angiogenesis genes Similar to Mesenchymal-like (M)	PI3K inhibitors Antiangiogenic therapy Src antagonist
Luminal androgen receptor (LAR)	Androgen receptor Luminal gene expression pattern Molecular apocrine subtype	Antiandrogen blockade CDK4/6 inhibitors Immune checkpoint inhibitors

Modified from Lehmann et al. [12,14] and Collignon et al. [27];AXL, tyrosine-protein kinase receptor UFO; CSC, cancer stem cells; EGFR, epidermal growth factor receptor; EMT, epithelial-mesenchymal-transition; FGFR, fibroblast growth factor receptors; IGF-1R, insulin-like growth factor receptor; IL, interleukin; MET, hepatocyte growth factor; mTOR,, mammalian target of rapamycin; NGF, nerve growth factor; PARP, poly ADP ribose polymerase; PDGFR, platelet-derived growth factor receptors; PD1, programmed cell death 1; PD-L1, programmed death-ligand 1; PI3K, phosphatidylinositol 3-kinase; TGFβ, transforming growth factor beta.

**Table 2 cancers-12-01404-t002:** Pathological complete response (pCR) rate in neoadjuvant trials with carboplatin in early stage TNBC.

Trials	Treatment	Number of Patients with TNBC	*pCR Rate	*p*-Value
GEICAM/2006–03 [40]	EC followed by T + carboplatin vs. without carboplatin	48 vs. 46	30% in both arms	N/A
GeparSixto GBG66 [36]	P and NPLD with Bev + carboplatin vs. without carboplatin	158 vs. 157	53.2% vs. 36.9%	0.005
GALGB 40603 Alliance [34]	(weekly) P + carboplatin, followed by ddAC (with or without Bev) vs. without carboplatin	221 vs. 212	54% vs. 41%	0.0029
Ando et al. [41]	(weekly) P + carboplatin followed by EC/5-FU vs. without carboplatin	37 vs. 38	61.2% vs. 26.3%	0.003
Zhang et al. [42]	P + carboplatin vs. P + E	47 vs. 44	38.6% vs. 14.0%	0.014
GeparOcto GBG84 [43]	(weekly) P and NPLD + carboplatin vs. E followed by P followed by C	203 vs. 200	51.7% vs. 48.5	0.518
WSG-ADAPT-TN [44]	Nab-P + carboplatin vs. Nab-P + G	154 vs. 182	45.9% vs. 28.7%	0.002
BrighTNess [45]	P + carboplatin followed by AC vs. without carboplatin	160 vs. 158	58% vs. 31%	0.0001
Sharma et al. [39]	T + carboplatin	190	55%	N/A
Yuan et al. [38]	Nab-P +carboplatin	67	48%	N/A

E: epirubicin; C: cyclophosphamide; T: docetaxel; P: paclitaxel; NPLD: non-pegylated liposomal doxorubicin; Bev: bevacizumab; A: doxorubicin; dd: dose-dense; G: gemcitabine; N/A: not applicable *pCR in the both breast and axilla (ypT0/is ypN0).

**Table 3 cancers-12-01404-t003:** Immune checkpoint inhibitor trials in early stage TNBC.

Trials	Treatment	Number of Patients with TNBC	*pCR Rate	*p*-Value
GeparNuevo [59]	Nab P + durvalumab followed by EC + durvalumab vs. without durvalumab	88 vs. 86	53.4% vs. 44.2%	0.287
KEYNOTE 173 [16]	Nab P with or without Cb + pembrolizumab followed by AC	60	60%	N/A
ISPY2 [57]	(weekly) P + pembrolizumab followed by AC vs. without pembrolizumab	29 vs.	60% vs. 22%	N/A
KEYNOTE 522 [15]	Cb/P + pembrolizumab followed by AC or EC + pembrolizumab vs. without pembrolizumab	401 vs. 201	64.8% vs. 51.2%	<0.001
NeoTRIP (2019, abstract) [60]	Cb and Nab P + atezolizumab vs. without atezolizumab	138 vs. 142	43.5% vs. 40.8%	0.66
NeoPACT (NCT03639948)	Phase II trial of Cb and T + pembrolizumab	Recruiting with accrual goal of 100	N/A	N/A
Impassion 031 (NCT 03197935) [62]	Phase III trial of Nab P + atezolizumab followed by ddAC + atezolizumab (continue atezolizumab as adjuvant after surgery)	Completed accrual with 204 patients	N/A	N/A
GeparDouze (NCT03281954) [61]	Phase III trial of (weekly) P and Cb + atezolizumab followed by AC or EC + atezolizumab	Recruiting with accrual goal of 1520	N/A	N/A

P: paclitaxel; Cb: carboplatin; A: doxorubicin; C: cyclophosphamide; E: epirubicin; T: docetaxel; dd: dose-dense; N/A: not applicable. *pCR in the both breast and axilla (ypT0/is ypN0).

**Table 4 cancers-12-01404-t004:** Poly-ADP-ribosyl polymerase (PARP) inhibitors in neoadjuvant TNBC trials.

Trials	Treatment	Number of Patients with TNBC	*pCR Rate	*p*-Value
I-SPY 2 [69]	P and Cb + veliparib followed by AC vs. P followed by AC	39 vs. 21	51% vs. 26%	Not reported (95% PI, 33–66% vs. 9–43%)
BrighTNess [45]	Arm 1: P and Cb + veliparib Arm 2: P and Cb Arm 3: P All arms followed by AC	316 vs. 169 vs. 58	53% vs. 58% vs. 31%	Arm 1 vs 2: 0.357 Arm 1 vs. 3: <0.0001
GeparOLA [71]	P+ olaparib vs. P + Cb, followed by EC	50 vs. 27	56.0% vs. 59.3%	Not reported
NCT02401347	Phase II of talazoparib	Recruiting with accrual goal of 40	N/A	N/A

P: paclitaxel; Cb: carboplatin; AC: doxorubicin and cyclophosphamide; EC: epirubicin and cyclophosphamide; N/A: not applicable. *pCR in the both breast and axilla (ypT0/is ypN0).

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
