# Peer review of "Neoadjuvant Treatment for Triple Negative Breast Cancer: Recent Progresses and Challenges"

_cancers, 2020, doi:10.3390/cancers12061404_

Round 1

Reviewer 1 Report

This review article summarizes the current trends of neoadjuvant therapy. The authors begin with TNBC molecular characteristics as well as the classification of differential TNBC subtypes, followed by NAT regimen, trial data, predictive biomarker, and novel approach. The structure is well thought out with detailed, updated, and appropriate trial data, which is highly beneficial to readers of the Cancers. I recommend this article, yet there are a few minor points that should be addressed before publication to be more informative for a broader audience.    

  1. Each treatment subcategory, the rationale of targeting cell type/signaling/pathway/molecules, are missing; I recommend to include explanations of why such drugs have been used for TNBC.
  2. Along the same line, I recommend to include MOA of drugs in the treatment section.
  3. In section 9, predictive biomarkers are discussed. In NAT therapy response marker, tumor types either pre/core, intra, post, and changes between pre and post therapy are considered. It should be explained what stage of tumors each study used.

Minor

  1. Line 106, n should be N, to be consistent
  2. Line 150, spell out EFS
  3. Line 246, alpha was lost in a conversion?
  4. Line 250, a symbol is lost
  5. Figure 1, this figure indicates TK (EGFR/HER) downstream signaling; however, given the TNBC focus, TK as upstream may be misleading. Especially, frequencies of PTEN mutation or aberrant activation of PI3K downstream are reported in TNBC. Additionally, PTEN “loss” may not fully capture TNBC disease etiology, and mutation should be included.
  6. Line 336, increased mRNA expression level of these genes reads both immune activating and suppressive genes. Please clarify if it means an increase in both gene sets or just immune activating genes.       

Author Response

Dear Reviewer, 

Thank you very much for your very helpful comments. We have completely revised our manuscript to respond to your comments.

Reviewer 2 Report

Dear authors, 

This manuscript is comprehensive, well-put-together, easy to read, that covers up-to-date agents of TNBC neoadjuvant therapy. 

The only comment I have is that the same Vanderbilt group who initially published JNCI paper in 2011, more recently came up with more narrowed 4 subgroups of TNBC instead of 6 listed here, and other groups also agree to these narrower subtypes. I strongly recommend the authors to either revise, or at least mention this newer categorization. 

Otherwise, good to go. Excellent job.

Author Response

Thank you very much for your very helpful comments. We have completely revised our manuscript to respond to your comments.
